# Investigation of the Reproducibility of Portable Optical Coherence Tomography in Diabetic Macular Edema

**DOI:** 10.3390/ph17101357

**Published:** 2024-10-11

**Authors:** Yoshiaki Chiku, Takao Hirano, Marie Nakamura, Yoshiaki Takahashi, Hideki Miyasaka, Ken Hoshiyama, Toshinori Murata

**Affiliations:** Department of Ophthalmology, School of Medicine, Shinshu University, 3-1-1 Asahi, Matsumoto 390-8621, Nagano, Japan

**Keywords:** diabetic macular edema, central macular thickness, portable OCT, intraretinal fluid, subretinal fluid, epiretinal membrane

## Abstract

**Background:** Diabetic macular edema (DME) causes vision impairment and significant vision loss. Portable optical coherence tomography (OCT) has the potential to enhance the accessibility and frequency of DME screening, facilitating early diagnosis and continuous monitoring. This study aimed to evaluate the reliability of a portable OCT device (ACT100) in assessing DME compared with a traditional stationary OCT device (Cirrus 5000 HD-OCT plus). **Methods:** This prospective clinical investigation included 40 eyes of 33 patients with DME. Participants with significant refractive errors (myopia > −6.0 diopters or hyperopia > +3.0 diopters), vitreous hemorrhage, tractional retinal detachment, or other ocular diseases affecting imaging were excluded. Spectral-domain OCT was performed by a single examiner using both devices to capture macular volume scans under mydriasis. Central macular thickness (CMT) was evaluated using the analysis software for each device: Cirrus used version 6.0.4, and ACT100 used version V20. We analyzed inter-evaluator and inter-instrument agreements for qualitative assessments of the intraretinal fluid (IRF), subretinal fluid (SRF), and epiretinal membrane (ERM) using Cohen’s kappa coefficient, whereas quantitative CMT assessments were correlated using Spearman’s correlation coefficient. **Results:** Substantial inter-evaluator agreement for IRF/SRF (κ = 0.801) and ERM (κ = 0.688) with ACT100 and inter-instrument agreement (κ = 0.756 for IRF/SRF, κ = 0.684 for ERM) were observed. CMT values measured using ACT100 were on average 29.6 μm lower than that of Cirrus (285.8 ± 56.6 vs. 315.4 ± 84.7 μm, *p* < 0.0001) but showed a strong correlation (R = 0.76, *p* < 0.0001). **Conclusions:** ACT100 portable OCT demonstrated high reliability for DME evaluations, comparable to that of stationary systems.

## 1. Introduction

Diabetic macular edema (DME) is a prevalent and potentially sight-threatening complication of diabetic retinopathy [1]. It is characterized by the accumulation of extracellular fluid in the macula, leading to vision impairment and significant vision loss [2]. DME affects a substantial proportion of patients with diabetes and is a major public health concern owing to the increasing global prevalence of diabetes [1].

The early and accurate detection of DME is crucial in preventing vision loss and initiating prompt and appropriate therapeutic interventions [3,4,5,6,7]. Optical coherence tomography (OCT) has emerged as the gold standard for diagnosing and monitoring DME because it provides high-resolution cross-sectional images of the retinal structures [8]. However, traditional OCT devices are often bulky and confined to clinical settings, which limits their accessibility, particularly in remote or underserved areas.

Recent advances in portable OCT technology have addressed these limitations. Compact and mobile OCT systems retain the high-resolution imaging capabilities of traditional OCT while offering flexibility for use in various settings, including primary care clinics, community health centers, and even patient homes. Thus, portable OCT has the potential to enhance the accessibility and frequency of DME screening, facilitating early diagnosis and continuous monitoring, especially in populations with limited access to specialized eye care services. Additionally, portable OCT devices are particularly beneficial for elderly patients and individuals with physical disabilities who cannot be transferred from wheelchairs to conventional stationary OCT chairs.

Portable OCT devices, such as ACT100 (Medimaging Integrated Solution Inc., Hsinchu, Taiwan), have shown promise in delivering reliable retinal measurements comparable to those obtained from traditional stationary OCT systems. Nakamura et al. investigated the reproducibility of the ACT100 portable OCT system and compared its performance with that of a conventional Cirrus HD-OCT (ZEISS, Jena, Germany) [9]. The study included 35 healthy participants and demonstrated that ACT100 provided consistent and reliable retinal thickness measurements. Despite the minor differences in the measured values, the intraclass correlation coefficients indicated high reliability, making ACT100 a viable alternative for clinical use. However, the validity of ACT100 has not yet been extensively verified in patients with retinal diseases. Further studies are needed to assess its performance in detecting retinal conditions, where structural changes and disease-related variability can pose challenges in yielding high-quality images, consistency, and measurement accuracy.

Therefore, this study aimed to evaluate the reliability of portable OCT in diagnosing DME. We compared the imaging outcomes of portable and traditional stationary OCT devices. Our objectives were to assess the agreement between these modalities and to determine the practicality of portable OCT in routine clinical practice and field settings.

## 2. Results

The study included 33 (26 male and 7 female) patients, with an average age of 64.1 ± 12.1 years. Of these patients, 26 had DME in only one eye, whereas 7 had DME in both eyes, resulting in 40 eyes being included in the analysis. The mean duration of diabetes was 15.5 ± 10.7 years, and the average HbA1c level was 7.4 ± 1.2%. The severity of diabetic retinopathy varied, with 2 eyes exhibiting mild non-proliferative diabetic retinopathy (NPDR), 8 eyes exhibiting moderate NPDR, 16 eyes exhibiting severe NPDR, and 14 eyes exhibiting proliferative diabetic retinopathy based on the International Clinical Diabetic Retinopathy Severity Scale [10]. Twenty-four eyes had undergone pan-retinal photocoagulation. The logMAR best-corrected visual acuity was 0.18 ± 0.27.

For the qualitative evaluation, we assessed the inter-evaluator and inter-instrument agreement for intraretinal fluid (IRF)/subretinal fluid (SRF) and epiretinal membrane (ERM) in all 40 eyes (Figure 1). Thirty-three out of the forty eyes (82.5%) showed agreement in all evaluations. For IRF/SRF, the Cohen’s kappa coefficient between the evaluators was 0.890 for Cirrus and 0.801 for ACT100. The inter-instrument agreement for IRF/SRF was 0.756. For ERM, the Cohen’s kappa coefficient between the evaluators was 0.875 for Cirrus and 0.688 for ACT100. The inter-instrument agreement for ERM was 0.684. Central macular thickness (CMT) was significantly lower for ACT100 (315.4 ± 84.7 μm for Cirrus and 285.8 ± 56.6 μm for ACT100) (Figure 2), but these measurements showed a correlation (R = 0.76, 95% confidence interval [CI] 0.59–0.87, *p* < 0.0001) (Figure 3).

## 3. Discussion

This study compared the reproducibility of portable OCT with that of existing stationary OCT in patients with DME. Our findings indicated that the CMT measured using ACT100, a portable OCT device, was on average 29.6 μm lower than that measured using Cirrus. The two devices showed a strong correlation, with an R-value of 0.76. When evaluating IRF, SRF, and ERM, the inter-evaluator and inter-instrument agreements were consistent in 82.5% of cases. The portability of ACT100, combined with its strong correlation with established devices, highlights its utility in diverse settings, particularly where conventional OCT systems are impractical.

Delayed medical visits by patients with DME had a significant impact, particularly during the coronavirus disease 2019 pandemic. A retrospective study showed that patients experienced an average delay of 2.4 months in their visits following the onset of the pandemic, with more than half of them exhibiting worsening DME [11]. Such delays underscore the urgent need for systems that facilitate the easier and more frequent evaluation of DME because untreated or the delayed treatment of DME can lead to irreversible vision loss.

Handheld fundus cameras are widely adopted as simple screening tools for diabetic retinopathy [12,13]. Screening for diabetic retinopathy using a mobile eye examination unit (EyeMo) decreased the rate of visual impairment by 86% [14]. Smartphone-based fundus screening is similar to a straightforward testing device [15]. However, the inability of these cameras to produce tomographic images poses a significant limitation, with false positive rates as high as 86.6% for the detection of DME [16].

Existing portable OCT devices have been developed to facilitate imaging in various settings, particularly for patients who cannot easily access traditional in-clinic OCT systems. The Heidelberg Spectralis with Flex Module, an armature-mounted system, was designed for use with supine patients [17]. However, its large size makes it less suitable for regular use at the bedside or in incubators [17]. The Envisu C-Class from Leica Microsystems is a commercially available handheld OCT system for supine patients. Despite its flexibility, the Envisu C-Class has a relatively low A-line scanning rate of 32 kHz, which compromises the image acquisition time and field of view and complicates the production of high-quality 3D volume images [18].

Recent technological advancements have significantly improved handheld OCT systems. Song et al. integrated faster scanning capabilities, real-time en face imaging, and on-probe displays to enhance operator experience and scanning efficiency, achieving faster 3D and wide-field imaging of the retina [19]. Additionally, ergonomic designs have been optimized by researchers like Viehland et al. for pediatric and supine imaging [20], and novel optical designs with advanced double aspheric lenses have been applied by Ni et al. to achieve non-contact handheld 105° ultrawide-field retinal imaging [21].

Despite these advancements, handheld OCT devices still face limitations, such as motion artifacts and lower imaging quality, compared with traditional table-mounted machines. Handheld OCT operations often induce motion between the probe and eye, causing spatial distortions and requiring multiple attempts to obtain satisfactory images, particularly in pediatric patients. Operators also experience fatigue and require substantial experience to consistently acquire high-quality images [20].

The new portable OCT ACT100 represents a significant advancement with its faster scan speed of 80 kHz, producing more detailed images and reducing motion artifacts. The ACT100 design incorporates an automatic focusing electric lens for easy alignment and an eyecup manipulator to minimize operator fatigue and motion artifacts. This system, which is placed on a portable cart, ensures easy movement and accessibility for bedside imaging, demonstrating the feasibility of non-contact high-resolution retinal imaging. However, the quality of the results depends on the operator’s experience. To maximize the device’s potential, clear guidelines on the level of training and experience are essential for ensuring acceptable and repeatable outcomes. Future studies should investigate the learning curve associated with ACT100, establishing the minimum required experience for optimal operation.

ACT100 may be helpful in screening for DME in remote islands and disaster areas. By enabling a more precise and timely evaluation, this device can help to carefully select patients who require anti-vascular endothelial growth factor (VEGF) treatment and facilitate their connection to secondary care facilities. This capability is crucial for ensuring that patients receive the necessary interventions without delay, which would lead to further deterioration of their vision. ACT100’s ease of use facilitates more frequent screening and monitoring in underserved regions, such as rural or low-population-density areas, where access to traditional stationary OCT systems is limited. Furthermore, ACT100’s mobility makes it ideal for deployment in disaster or field settings, where rapid responses and equipment setup are critical. These features underscore ACT100’s potential to enhance the much-needed access to eye care in various challenging environments.

In the era of anti-VEGF therapy for DME, the central retinal thickness, along with visual acuity, is one of the most critical quantitative metrics used to make treatment decisions in most clinical trials [22]. However, it is important to note that variations in CMT measurements between different OCT devices, such as Cirrus and Spectralis (Heidelberg Engineering, Heidelberg, Germany), have been documented [23,24]. The central subfield thickness (CST) on Spectralis was greater than the CST on Cirrus by a mean difference of 21 µm [23]. These variations can be attributed to differences in the algorithms used for measuring the CST, and particularly in how the boundaries for CST are defined [25].

Nakamura et al. validated ACT100 by examining the CMT in normal eyes, finding that ACT100 measured the CMT to be on average 10 μm smaller compared with that measured by Cirrus [9]. Both devices measured from the internal limiting membrane to the center of the retinal pigment epithelium, and no difference in the measurement range was observed between the two models. Thus, the potential causes of the differences in CMT measurements include variations in scan spacing, refractive index, and longitudinal resolution [25].

In our study, the CMT measured using ACT100 was on average 29.6 μm lower than that measured using Cirrus. Both Cirrus and ACT100 used 512 A-scans, and the number of B-scans was almost identical, with Cirrus at 128 and ACT100 at 129. However, the measurement range differed, with Cirrus covering a 6.0 × 6.0 mm area and ACT100 covering a 9.0 × 9.0 mm area. This broader range for ACT100 resulted in wider scan intervals than those for Cirrus, contributing to differences in the measured values. Additionally, the refractive index for converting the optical delay in ACT100 was 1.379, a value not reported for Cirrus. Differences in the refractive indices between models can also lead to measurement discrepancies. In clinical settings, this can be managed by applying conversion factors for cross-device comparisons and ensuring that clinicians are trained to account for these differences when interpreting results. Furthermore, the longitudinal resolution varies significantly between ACT100 (10 μm) and Cirrus (5 μm), which can impact measurement accuracy.

The discrepancy in CMT values observed in our study was greater than that reported in previous studies of normal eyes [9]. This large measurement discrepancy could be attributed to additional factors. First, the mean CMT value was higher in patients with DME than that in normal eyes. Therefore, if the measurement error is proportional, the absolute error would be larger. In this study, the CMT was 315.4 μm on Cirrus and 285.8 μm on ACT100. The previous study reported that the CMT in the right eye was 257.9 μm on Cirrus and 245.8 μm on ACT100 in normal eyes, indicating a 22% and 16% increase in DME eyes in each model, respectively. However, this proportional error does not fully account for the observed increase in mean CMT values compared with normal eyes. Other factors contributing to measurement discrepancies include the challenge of accurately identifying the fovea in patients with DME, potentially leading to variations in the measurement position and increased motion artifacts owing to poor fixation and eye movements. It is important to recognize that CMT measurement errors may be larger in diseased eyes. Caution is necessary when comparing CMT values obtained from different OCT devices. Variations in measurements can affect clinical decision-making and the interpretation of research outcomes. Thus, when using different OCT devices interchangeably, it is essential to consider these differences and apply appropriate conversion equations to ensure the consistency and accuracy of CMT measurements.

The findings of this study on the inter-evaluator agreement of OCT variables using a portable OCT device have significant implications for the clinical management of DME. We observed a high inter-evaluator reliability for IRF/SRF (κ = 0.756) and ERM (κ = 0.684) in portable OCT. These values are comparable to those reported in the existing literature, where kappa values for IRF/SRF were 0.82 and 0.85, respectively, while, for ERM, the value was 0.37 [26].

Our findings indicated that the CMT measured using ACT100 was on average 29.6 µm smaller than that measured using Cirrus. In clinical practice, CMT is a key metric for diagnosing and monitoring DME and plays a critical role in guiding treatment decisions, especially for anti-VEGF therapy administration. A deviation of 29.6 µm, although statistically significant, may not always necessitate different treatment decisions in all cases. However, such differences become clinically relevant in borderline cases, where even slight variations in CMT measurements can affect treatment thresholds. To address this, the use of correction factors or adjustments may be necessary when using ACT100 in clinical settings. Monitoring trends in CMT over time, rather than relying on a single measurement, may also help mitigate the impact of these discrepancies. Furthermore, incorporating device-specific adjustments, such as correction factors, could standardize results across different OCT systems.

Recent advancements in OCT technology, such as full-range optical coherence refraction tomography (FROCRT), offer enhanced depth resolution and eliminate mirror artifacts by capturing complex conjugate-free images from multiple angles. FROCRT improves axial resolution and extends imaging depth, overcoming the limitations of traditional OCT [27]. Applying such innovations to portable OCT devices such as ACT100 could further improve diagnostic accuracy, particularly in retinal disease assessments. Future studies must explore integrating FROCRT to enhance the precision of portable OCT systems in measuring parameters such as CMT.

This study has several limitations. First, the sample size was relatively small (40 cases). Future studies should include larger sample sizes and expand the focus beyond DME to other retinal diseases, such as age-related macular degeneration and retinal vein occlusion. However, similar sample sizes have been successfully used in previous studies of DME [28,29], supporting the relevance of our findings even with this sample size. Additionally, caution must be exercised when interpreting the results of portable OCT because its image quality can be affected by motion artifacts and the skill level of the examiner. However, ACT100 has features such as an eyecup for probe stabilization and an automatic alignment function, which can help achieve clinically satisfactory imaging proficiency with only a few hours of training. Despite its high-speed A-scan of 80 kHz, ACT100 is still inferior to the existing OCT systems in terms of resolution, indicating the need for further improvements. As these challenges are commonly associated with portable OCT devices, future studies should compare ACT100 with other portable devices, such as the Heidelberg Spectralis Flex Module (Heidelberg Engineering, Germany) and Envisu C-Class (Leica Microsystems, Wetzlar, Germany). Such comparisons would offer a more sophisticated analysis of ACT100’s strengths and weaknesses, especially in terms of resolution and motion artifact handling, and help better position it within the range of portable retinal imaging tools. Another potential application area is retinopathy owing to prematurity, although it remains to be determined whether portable OCT can acquire high-quality images in premature infants. Further studies are required to explore these potential applications. Although participants with significant refractive errors were excluded, mild refractive errors were not accounted for in the analysis. Future investigations should assess the impact of mild to moderate refractive errors on the accuracy of portable OCT measurements. The absence of a control group with normal eyes was another limitation. Including such a group in future research would provide a clearer understanding of the discrepancies in OCT measurements between healthy and DME-affected eyes.

## 4. Materials and Methods

This study was conducted on patients with DME between 8 and 26 May 2023. This study was approved by the Ethics Committee of Shinshu University (approval no. 5923). All procedures were performed in accordance with the principles outlined in the Declaration of Helsinki. Written informed consent was obtained from all participants. The participants were recruited from the Department of Ophthalmology at the Shinshu University Hospital. Participants were excluded if any of the following were present: a significant refractive error (myopia > −6.0 diopters or hyperopia > +3.0 diopters), vitreous hemorrhage, tractional retinal detachment, and ocular diseases that cause poor imaging, such as corneal opacity. OCT imaging was performed using the two instruments on each patient on the same day by a single examiner familiar with the testing method. Imaging was performed by obtaining volume scans centered on the macula under mydriasis using two OCT systems: a portable spectral-domain (SD)-OCT system (ACT100 ver.19, MiiS, Hsinchu, Taiwan) and an existing stationary SD-OCT system (Cirrus 5000 HD-OCT plus, Carl Zeiss, Baden-Württemberg, Germany). Images were captured until sufficient confidence counts were achieved, with a signal index of 6 or higher for ACT100 and a signal strength of 6 or higher for Cirrus. Up to three scans were allowed and patients were excluded if adequate images could not be obtained within this limit.

CMT was evaluated using an average value of the inner 1 mm diameter within the Early Treatment Diabetic Retinopathy Study circle, which was automatically evaluated using the analysis software of each device: Cirrus used version 6.0.4, and ACT100 used version V20 [30]. The measurement protocols used for each device were as follows.

Cirrus: 6.0 × 6.0 mm area captured with a macular cube scan. Each image comprised 512 A-scans × 128 horizontal scans.

ACT100: 9.0 × 9.0 mm area captured with a thickness map retina scan. Each image comprised 512 A-scans × 129 horizontal scans.

Size and weight of the two OCT models were as follows: stationary Cirrus (46.0 cm × 65.0 cm × 53.0 cm, 36 kg) and ACT100, a portable OCT with a main box (23.4 cm × 19.0 cm × 10.8 cm, 3.35 kg) and probe (27.5 cm × 9.0 cm × 7.4 cm, 0.75 kg) (Figure 4). Both models were spectral-domain OCTs with a light source wavelength of 840 nm. The scanning speeds were 27,000–68,000 A-scans/s for Cirrus and 80,000 A-scans/s for ACT100. The transverse resolution was 15 µm for Cirrus and 20 µm for ACT100, with longitudinal resolutions of 5 µm and 10 µm, respectively. The acquisition times were 1.5 s for Cirrus and 2.5 s for ACT100.

The acquired OCT images were qualitatively evaluated for the presence of IRF/SRF and ERM. The presence of IRF/SRF was evaluated in four groups: IRF only, SRF only, presence of both IRF and SRF, and absence of both IRF and SRF. The central B-scan image was evaluated vertically and horizontally to examine the presence or absence of IRF/SRF and ERM (Figure 5). Only contrast adjustment was allowed during the evaluation, and no image enlargement, reduction, or color-tone changes were permitted. Two retinal specialists evaluated the images individually, and their initial agreements were compared for inter-evaluator assessment. Any discrepancies between their assessments were discussed to reach a final consensus. Subsequently, the inter-instrument agreement between the two OCT models was assessed based on their final evaluations.

### Statistical Analysis

All statistical analyses were performed using IBM Statistical Package for the Social Sciences (SPSS) version 22.0 (IBM Corp., Armonk, NY, USA). Continuous variables were reported as mean values with standard deviations. Agreement was assessed using Cohen’s kappa coefficient. Spearman’s correlation coefficient was used to evaluate the correlation between the two models. Due to the small sample size and the inability to assume normal distribution, Spearman’s rank correlation coefficient was used for statistical analysis instead of the intraclass correlation coefficient, which requires normally distributed data. Statistical significance was set at *p* < 0.05.

## 5. Conclusions

This study demonstrated that ACT100 portable OCT offers high reliability for both qualitative and quantitative evaluations of DME, comparable to that of traditional stationary OCT systems. ACT100 showed substantial inter-evaluator and inter-instrument agreement in assessing IRF, SRF, and ERM. Despite the measured CMT values being slightly lower than those of Cirrus, the strong correlation between the two devices underscores the accuracy of ACT100. The portability and ease of use of ACT100 enhance its applicability in diverse clinical settings, including remote and underserved areas, where access to stationary OCT systems is limited. Although the sample size of this study was limited to 40 eyes from 33 patients, its findings provide valuable initial insights. Future studies with larger cohorts will be necessary to confirm the reproducibility and clinical applicability of ACT100 portable OCT.

## Figures and Tables

**Figure 1 pharmaceuticals-17-01357-f001:**
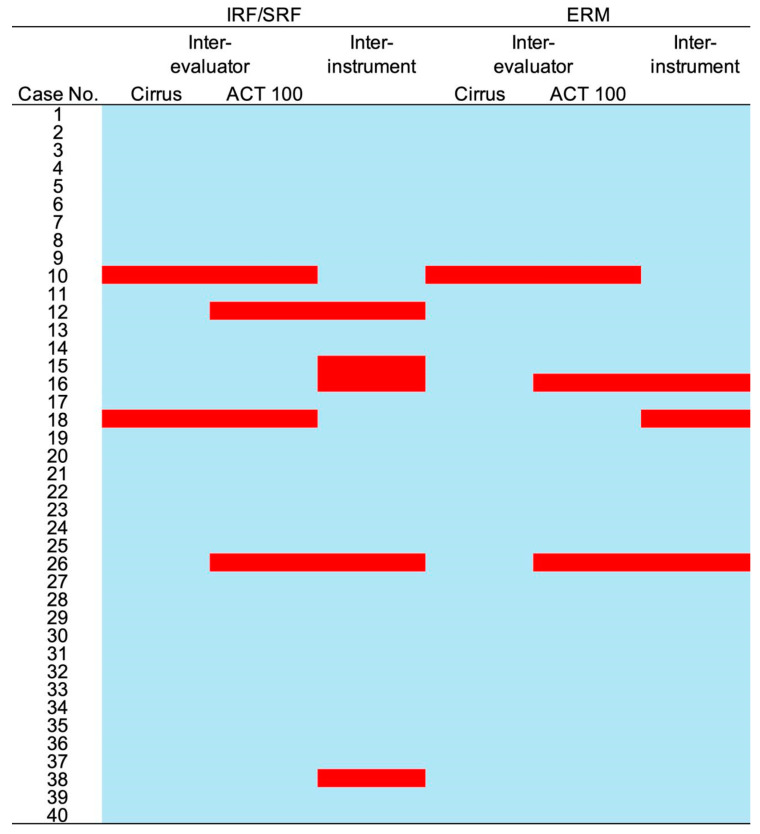
Inter-evaluator and inter-instrument agreement in OCT evaluations. A qualitative evaluation assessed the inter-evaluator and inter-instrument agreement for the presence of intraretinal fluid (IRF), subretinal fluid (SRF), and epiretinal membrane (ERM) in all 40 eyes. Blue highlights indicate agreement between the evaluators and instruments, whereas red highlights indicate discrepancies. Out of 40 eyes, 33 eyes (82.5%) showed agreement in all evaluations.

**Figure 2 pharmaceuticals-17-01357-f002:**
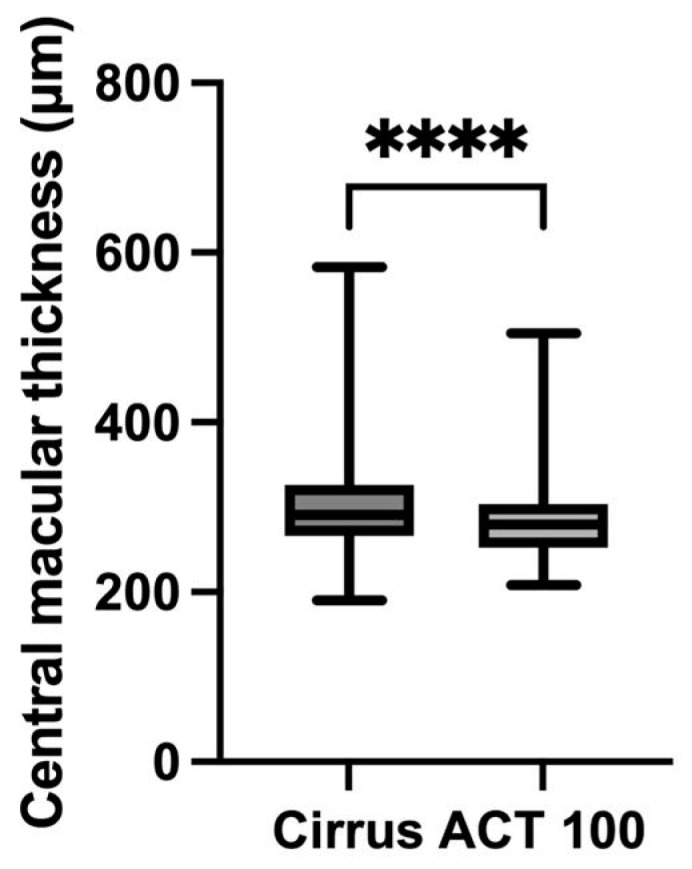
Comparison of central macular thickness (CMT) between Cirrus and ACT100. CMT measured with Cirrus was 315.4 ± 84.7 μm, whereas that measured with ACT100 was 285.8 ± 56.6 μm (**** *p* < 0.0001, Wilcoxon signed-rank test).

**Figure 3 pharmaceuticals-17-01357-f003:**
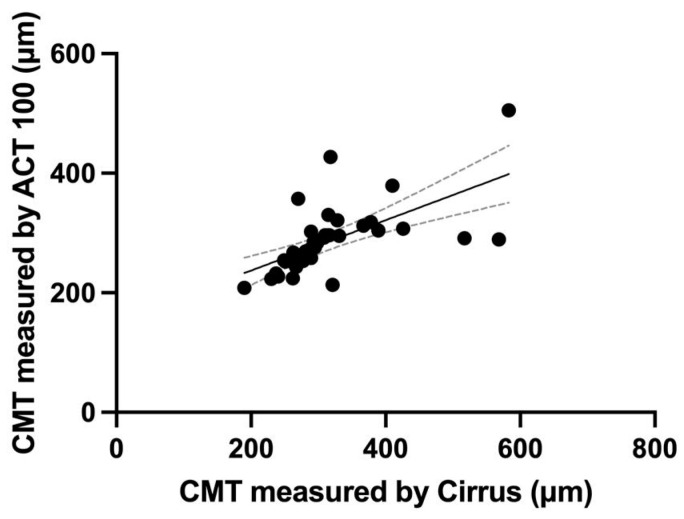
Correlation of central macular thickness (CMT) measurements between Cirrus and ACT100. The correlation between CMT measurements obtained from the Cirrus and ACT100 devices was R = 0.76 (95% confidence interval 0.59–0.87, *p* < 0.0001 in Spearman’s correlation coefficient).

**Figure 4 pharmaceuticals-17-01357-f004:**
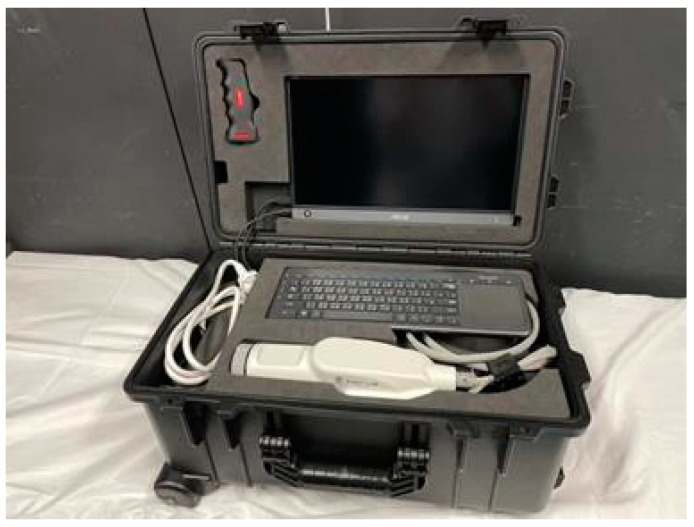
ACT100, a portable OCT system with a carrying case. It is a compact and lightweight portable OCT system comprising a main unit measuring 234 × 190 × 108 mm and weighing 3.35 kg, along with a probe measuring 275 × 90 × 74 mm and weighing 0.75 kg. The system’s accessories, including cables and monitors, fit into a single carrying case. This case measures 290 × 280 × 520 mm, weighs 15.4 kg, and is designed to be portable like a standard rolling suitcase.

**Figure 5 pharmaceuticals-17-01357-f005:**
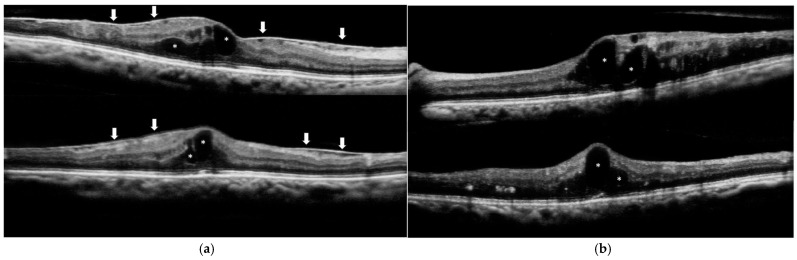
Vertical and horizontal OCT images from ACT100 of patients with diabetic macular edema. The upper image is a horizontal scan, and the lower image is a vertical scan in both panels. (**a**) The presence of intraretinal fluid (IRF) (white asterisk) and epiretinal membrane (white arrow) in a patient. (**b**) The presence of IRF (white asterisk) in another patient.

## Data Availability

The data supporting the findings of this study are available from the corresponding author upon reasonable request. The data are not publicly available due to privacy concerns, as the data involve sensitive patient information.

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
