# Peer review of "Investigation of the Reproducibility of Portable Optical Coherence Tomography in Diabetic Macular Edema"

_pharmaceuticals, 2024, doi:10.3390/ph17101357_

Round 1
Reviewer 1 Report
Comments and Suggestions for Authors
Dear authors,
The manuscript " Investigation of the reproducibility of portable optical coherence tomography in diabetic macular edema" is an interesting study and would be very interesting for readers. However I feel the overall presentation of the manuscript can be improved if you please address the following questions.
Although it is worth noting that this study only includes 30 eyes of 23 patients, which is rather a small sample size, the sample size though increased would have made the findings more rigorous and the results more applicable to the general population.
It is stated in the text that participants with high refractive errors were not included in the study. However, the effect that mild refractive errors had on the results was never addressed. It would also be interesting to know whether or not moderate refractive errors could have impacted the analysis.
It is stated in the study that ACT100 was able to measure central macular thickness at an average of 36.4 μm lesser CMT compared to Cirrus OCT on CMT. The difference, while statistically significant, has not been exhaustively expounded upon regarding the practical implications of this difference which is not very common. The difference in measurements made and how those differences in decisions made by the physician could be through education of the physician on these limitations and how best to address deviations/calculate outliers.
Similar results were observed regarding agreement among instruments for measurements of CMT though the agreement was good but never absolute. Such measures should be taken by the authors and even more so enhanced in a clinical setting to explain how all such differences could be accounted for perhaps as a correction factor or adjustment but used during usage with ACT100 device.
The omission of a control group consisting of subjects with normal eyes is a weakness of the study. More specifically, this would provide the ground for the comparison and demonstration of whether the discrepancies observed are limited to the diabetic macular edema affecting eyes only.
The authors concede the issues regarding duality of the resolution and the possible motion defects of the ACT100 device. Nevertheless, the article lacks such comparisons with other portable for conducting a more sophisticated analysis of the obtained results.
It is reported that the ACT100 device is designed in a way to allow the user to operate it with very little training. It should be noted however that, a time and experience limits on the operator's part for acceptable repeatable results should be defined since this could limit the uptake of the device.
The focus of the present work is on the compactness of the ACT100 device however, the specific clinical situations where this device would have superiority has not been addressed. More detailed discussion of the possible usage of the device in the field, for example in the areas with low population density or other underserved areas, and would increase the quality of the paper.
Cohen's kappa index and Spearman's rank order coefficient of correlation apply very well for the purposes of the current study. However, it would be reasonable to include in the paper a more detailed justification for the selection of these statistical methods instead of others, such as the use of infraclass correlation coefficients, popular in reproducibility studies.
Some differences in refractive indices between the two devices, ACT100, and Cirrus OCT are referred to as a reason for possible discrepancies in the measurements. It may be beneficial for the readers to include more detail about the management of these types of refractive index differences in clinical settings.
The paper states that large sample sizes should be made to address these issues in the future. It would be helpful to mention what particular future studies suggest, for example, other retinal diseases or the use of another more convenient portable OCT for the comparison studies in this area.
The similarity index of the article is bit too high, I request the authors to please reduce it.
best regards,
Comments on the Quality of English Language
Minor editing of English language required.
Reviewer 2 Report
Comments and Suggestions for Authors
The objective of this study is to evaluate the reliability of a portable OCT device (ACT100) in assessing DME compared with a traditional stationary OCT device (Cirrus 5000 HD-OCT plus). The study observations suggest that ACT100 portable OCT demonstrated high reliability for DME evaluation, comparable to that of stationary systems. If these following problems (in the attached file) can be solved, this research has the potential to be published in the journal MDPI Pharmaceuticals.

Round 2
Reviewer 2 Report
Comments and Suggestions for Authors
Most of the replies satisfy me, but the authors have to cite more related references in their manuscript.
